# COVID-19 Vaccine Hesitancy: The Role of Information Sources and Beliefs in Dutch Adults

**DOI:** 10.3390/ijerph19063205

**Published:** 2022-03-09

**Authors:** Hein de Vries, Wouter Verputten, Christian Preissner, Gerjo Kok

**Affiliations:** 1Department of Health Promotion, Faculty of Health, Medicine and Life Sciences, Maastricht University, 6200 MD Maastricht, The Netherlands; w.verputten@student.maastrichtuniversity.nl (W.V.); c.preissner@maastrichtuniversity.nl (C.P.); 2CAPHRI School for Public Health and Primary Care, Faculty of Health, Medicine and Life Sciences, Maastricht University, 6200 MD Maastricht, The Netherlands; 3School of Psychology & Neuroscience, Maastricht University, 6200 MD Maastricht, The Netherlands; g.kok@maastrichtuniversity.nl

**Keywords:** COVID-19, vaccine hesitancy, knowledge, attitudes, self-efficacy, information seeking behaviour, I-Change Model

## Abstract

COVID-19 vaccine hesitancy may be regarded as a new pandemic hindering the elimination of or coping with COVID-19. This study assessed reasons for COVID-19 vaccine hesitancy using the I-Change Model (ICM) by considering the role of informational and psychosocial factors. A cross-sectional online survey using a convenience sample was conducted among Dutch adults (*n* = 240). The questionnaire assessed information factors, predisposing factors, awareness factors, motivational factors, preparatory actions, and vaccination intention. Vaccine hesitant participants (*n* = 58, 24%) had lower levels of education, more often paid work, and tended to have a religion other than Catholicism. They used written media less often and tended to visit websites of public health organizations less often, but used messaging services like WhatsApp more frequently. All participants had neutral intentions towards checking information credibility. Vaccine hesitant respondents had less knowledge about vaccination, lower perceived severity of getting sick and dying of COVID-19, and reported fewer exposures to cues about the advantages of COVID-19 vaccination. They were less convinced of the emotional and rational advantages of COVID-19 vaccination and expressed more negative feelings about it. They also reported more negative social norms concerning COVID-19 vaccination, and lower self-efficacy to get vaccinated and to cope with potential side-effects. The regression model explained 58% of the variance in vaccination intention. The results suggest that strategies are needed to: 1. Reduce fake news and stimulate information checking to foster well-informed decision-making; 2. Target both rational and emotional consequences of COVID-19, in addition to strategies for optimizing levels of knowledge. Campaigns should acknowledge the perceptions of the emotional disadvantages and increase perceptions of emotional advantages of COVID-19 vaccinations, such as reducing feelings of regret, and increasing feelings of freedom and reassurance.

## 1. Introduction

The COVID-19 pandemic is increasingly resulting in high levels of mortality and morbidity. Many effective vaccines have been approved for use [1], and a relatively high vaccination level in the Netherlands has been realized, resulting in full vaccination of 84% in the Dutch population of 12 years and older [2]. Yet, COVID-19 keeps resulting in new waves of mortality and morbidity due to new mutations and vaccine hesitancy [2,3,4,5]. Vaccine hesitancy (VH) can be described as a person’s uncertainty about whether to get vaccinated or not [2].

One of the factors influencing VH includes the sources and types of information that serve as cues for people to think and act with respect to the COVID-19 vaccination. Exposure to social media has been shown to be an important cue for people to become aware of COVID-19 and of actions relevant to cope with it [6]. An important characteristic of social media concerns the development of information bubbles [6], which are particular spheres of information that persons have access to resulting from algorithms selecting information based on prior content interactions of the user. This may lead to people seeing information that confirms what they already believe in [7]. Additionally, the online environment also contains so-called fake news, which may heavily influence COVID-19 VH [8]. Fake news, or misinformation, is information that cannot be verified or is not evidence-based [9]. The increasing uncertainty about the trustworthiness and credibility of news sources is problematic, as it may encourage an increase in VH and decreased trust in the government [10]. Previous research suggests that people may be more susceptible to fake news, and therefore VH, if they do not know how to properly navigate the Internet [11]. In this regard, taking preparatory actions against misinformation may help people to lower the impact of fake news on the Internet. Social media plays a vital role in the creation and spread of fake news, as information free of restrictions can easily be spread. A recent investigation of fake news on social media showed that 80% of it is created by 0.1% of all accounts [12]. This study also suggested that 80% of that news is consumed by 1% of all accounts [12], which may be problematic for VH when the affected individuals spread the incorrect information in their social environment. This is known as second-hand misinforming [12].

Research indicates that higher educated adults gathered health information from various reliable sources, whereas adults that received a lower education were more likely to use family and friends as a source for health information [13], increasing the chance of second-hand misinforming. Consequently, the spread of fake news via social media and interpersonal communication may influence various individual and social determinants of VH. Among the personal determinants, knowledge [14], risk perception [15], and attitudes toward vaccination [16] were associated with COVID-19 VH. Furthermore, a review on the human papillomavirus (HPV) vaccine identified ten thematic categories of VH determinants [17]. The emerging determinants across these categories were risk perception, perceived advantages and disadvantages (i.e., attitudes), knowledge, and perceived social norms [17]. Hence, to optimize communication and to better reach people that are hesitant towards COVID-19 vaccination, targeted approaches are needed that address the relevant beliefs about it.

This study uses the I-Change Model (ICM) to examine the influence of informational and personal motivational factors on COVID-19 VH [18]. The ICM suggests that awareness, motivation, and action concerning health promoting behaviours can be explained by information factors (e.g., choice of Internet sources, message quality) and predisposing factors (e.g., gender, age, education) (see Figure 1). The awareness of a particular health issue (e.g., the need for COVID-19 vaccination) is suggested to be determined by knowledge (about the vaccination), perceived cues (events that guide behaviour), risk perception (a person’s perceived severity of a health threat and their perceived susceptibility to the threat) and cognizance (a person’s awareness of their own level of health behaviour). These awareness factors influence motivational factors such as a person’s attitude (towards vaccination), social norms and support (towards vaccination), and self-efficacy (the confidence a person has to overcome barriers that may hinder a health promoting behaviour, such as vaccination) [19]. These factors determine a person’s level of motivation towards health promoting behavior or their intention to engage in it. Translations from intention to behaviour can be facilitated by preparatory planning, i.e., choosing relevant action plans, and coping plans to overcome barriers that may hinder a person’s action plans [19]. The relevance of these post-intentional constructs has previously been demonstrated for health behaviours such as inspecting risks for hereditary cancer [20], physical activity [21], and children’s vaccination uptake by parents [22].

The goal of this study is to analyse the role of informational and motivational factors associated with VH against COVID-19. Using the ICM, the objectives are to examine adults differing in COVID-19 VH concerning their (i) sociodemographic factors, (ii) information seeking behaviour, including actions to check information and source credibility, (iii) awareness, and (iv) motivational factors regarding the COVID-19 vaccine. This in-depth information may help to optimize communication about COVID-19 vaccination for this group. Specifically, it may aid in the development of tailored approaches that also aim at understanding VH rather than merely focusing on persuasion.

## 2. Materials and Methods

A cross-sectional study was conducted in line with the ICM to measure vaccination intention, preparatory actions to combat fake news, information seeking behaviour, awareness factors, and motivational factors. The anonymized dataset is available upon request from the first author.

### 2.1. Design

The study used a quantitative method to gather data about psychosocial determinants that might have an association with vaccination intention. Data collection for this study took place online from April to May 2021. Before filling in the questionnaire, respondents were informed about the purpose of the study. Informed consent was obtained from all individuals involved in the study through an informed consent form at the start of the online questionnaire.

### 2.2. Study Population

This study used a convenience sample and focused on Dutch-speaking adults, aged 18+, as they were the target population for the vaccination campaign against COVID-19 in the Netherlands in which four types of vaccinations were used (Pfizer, Moderna, AstraZenica, Johnson & Johnson); allocation of vaccines was decided top-down depending on the age of the person.

The link to the survey was disseminated to the target population online via WhatsApp, LinkedIn, and Facebook. Participants were required to (i) be above 18 years of age and (ii) not have been vaccinated yet. Of the original 325 respondents, 40 were excluded because they indicated that they were partially or fully vaccinated, 39 because they did not fully complete the questionnaire (<90% finished), and six because their age was lower than 18. The anonymized data of the remaining 240 respondents were included in the study.

### 2.3. Questionnaire

Questionnaire development was informed by previous research using the I-Change Model (see, e.g., [18,19,20,21,23,24]). The questionnaire was piloted for comprehension and content by individuals from the target group and adjusted according to participants’ feedback. These participants were not eligible to fill in the final online questionnaire on Qualtrics. The average time to complete the questionnaire was 15 min.

#### 2.3.1. Vaccine Hesitancy

The intention to get vaccinated was measured on a 3-point Likert scale (−1 = surely not, 0 = I don’t know, 1 = surely) [25] using seven items (α = 0.93), assessing a general vaccination intention (3 items; α = 0.97) and a COVID-19 vaccine-specific intention (4 items; α = 0.87). Low scores reflect greater VH.

#### 2.3.2. Socio-Demographic Factors

Socio-demographic factors were assessed using questions from the Dutch Health Monitor [24]. Participants were asked to indicate their age, their gender (1 = male, 2 = female, 3 = other), their highest completed education (1 = no education, 2 = primary school, 3 = lower vocational education, 4 = lower level secondary school, 5 = middle vocational education, 6 = higher level secondary school, 7 = higher vocational education, 8 = scientific education), their working situation (1 = working 1–11 h per week, 2 = working 12–19 h per week, 3 = working 20–31 h per week, 4 = working 32 or more h per week, 5 = retired, 6 = unemployed, 7 = disability pension, 8 = receiving financial aid from the government, 9 = being a homemaker, 10 = student) (more than one answer could apply) and their religion (0 = not religious, 1 = Protestant, 2 = Catholic, 3 = Muslim, 4 = Anthroposophist, 5 = other; for the regression analysis the coding was: 0 = not religious, 1 = Catholic; 2 = other religions) (see Table 1 for more details).

#### 2.3.3. Information Seeking Behaviour and Quality Checking Intentions

Information seeking behaviour was assessed by 12 questions (see Table 2 for the items) about the frequency with which someone used a prespecified media or interpersonal source (0 = never, 1 = very rarely, 2 = rarely, 3 = sometimes, 4 = often, 5 = very often). The list of sources was based on previous studies about information seeking behaviour [26,27,28].

Intentions for information quality checking were assessed using a 7-point Likert scale (−3 = surely not to +3 = surely yes) to assess whether the respondent had the intention to: (1) check whether the news they read was fake or not; (2) read sufficient information on websites or social media accounts of the government or public health organisations; (3) read sufficient information on other websites or social media accounts; (4) obtain sufficient information from their family and friends; (5) check if their information from websites or social media accounts of the government or public health organisations is reliable; (6) check if their information from other websites or social media accounts is reliable; (7) check if the information from family and friends is reliable; (8) check the correctness of the information from social media accounts of the government or public health organisations; (9) check the correctness of information of other websites or social media accounts; and (10) check the correctness of information from family and friends.

#### 2.3.4. Awareness Factors

Knowledge about the facts concerning COVID-19 vaccination was measured using seven items regarding the vaccine (0 = incorrect, 1 = correct; α = 0.56). Perceived susceptibility (α = 0.78) and perceived severity towards getting COVID-19 (α = 0.64) were both measured with four items on a seven-point Likert scale (−3 = extremely low to 3 = extremely high). Cues to action were measured by five questions asking about possible cues for vaccination (0 = no, 1 = yes; α = 0.44) [29]. The low α’s for knowledge and cues were not considered as problematic, as these scales did not measure one dimension and thus serve as an index.

#### 2.3.5. Motivational Factors

Attitude, self-efficacy, social norms, and intention were measured on 7-point Likert scales (−3 = completely disagree to +3 = completely agree). An overview of the items can be found in Table 3.

Attitude beliefs were assessed as both rational and emotional outcome expectations [18]. Potential advantages of COVID-19 vaccinations included seven rational consequences (α = 0.89) and six emotional consequences (α = 0.86). Potential disadvantages of COVID-19 vaccination encompassed six rational outcomes (α = 0.79) and five emotional outcomes (α = 0.91).

Social norms towards getting vaccinated against COVID-19 were assessed using seven items on a 7-point Likert scale (α = 0.91). An additional response option was provided in case a specific item did not apply (n/a = missing value = 0).

Self-efficacy was assessed using a 7-point Likert scale for 12 items (α = 0.91). Questions were developed based on previous studies. These items were reverse scored, i.e., a higher score reflects a higher self-efficacy towards getting vaccinated.

### 2.4. Data Analysis

Data was analysed using SPSS version 27 (SPSS Inc., Chicago, IL, USA). Descriptive analyses were used to describe the research population. Missing data only occurred in five individuals for preparatory actions and was not replaced. Participants were divided into three groups corresponding to their level of VH. To investigate differences between the hesitancy groups on psychosocial constructs of the ICM, a summary measure of intention for the specific vaccines was computed (α = 0.87). The non-hesitant group had an intention score of >0.99, the somewhat hesitant group had an intention score of 0.50–0.99 and the hesitant group had an intention score of <0.50. These cut-off points were based on the distribution of the mean of the vaccine-specific intention summary measure. Approximately half of the population was non-hesitant (*n* = 113; 47.1%), the rest was somewhat hesitant (*n* = 69; 28.8%) or hesitant (*n* = 58; 24.2 %).

Differences between the groups in categorical data were measured using chi-square tests. Differences between the groups for interval data were measured by one-way ANOVAs. Bonferroni-adjusted post-hoc tests were used to determine which groups significantly differed from each other. Finally, a multiple linear regression analysis was conducted (method: Forward) to determine factors uniquely associated with vaccination intention and to identify how well the model explained the variance in intention.

## 3. Results

### 3.1. Respondent Characteristics

Table 1 displays respondent characteristics. The study population was predominantly female (75.0%), with a mean age of 44.1 years, higher education (an average education level of 6.5 on a scale of 8), and mostly employed (72.1%). Most of the respondents were either not religious or Catholic.

### 3.2. Differences in Socio-Demographic Factors

The data presented in Table 1 showed no differences for age and gender between the VH groups. Non-hesitant and somewhat hesitant respondents reported to have completed significantly higher levels of education than hesitant respondents. Hesitant and somewhat hesitant participants more often reported to have paid work than non-hesitant participants. A trend suggested that the percentage of hesitant and somewhat hesitant participants was highest among individuals with a religion other than Catholicism.

### 3.3. Differences in Information Seeking Behaviour

Table 2 shows that written media was used significantly less often by hesitant respondents than the other two groups as the news source for information regarding COVID-19 vaccination. We found an additional trend suggesting that hesitant individuals used social media accounts or websites of public health organisations less often than the other two groups. On the other hand, hesitant respondents used messaging services like WhatsApp more often than the other two groups. Yet, the latter two differences were not found in the contrast analysis, potentially due to a relatively low level of respondents in the two hesitant groups. Concerning the 10 items for information checking intentions, no differences were found for the three groups. All means were around 0, revealing no significant intentions to check the quality of information for these 10 sources.

### 3.4. Differences in Awareness

Table 3 shows that hesitant respondents had less knowledge than somewhat hesitant and non-hesitant respondents concerning possible infection after vaccination, side-effects, spread of the virus after vaccination, how extensively the vaccines have been tested, and the safety of the vaccines.

Concerning risk perception, respondents all had moderate perceptions concerning their chances of getting COVID-19. The somewhat hesitant respondents showed the lowest perception of getting seriously sick and of experiencing permanent damage, which significantly differed from the non-hesitant group. Regarding the perceived severity, hesitant respondents showed significantly lower perceived severity concerning getting seriously sick from COVID-19 and of dying of COVID-19 than the other respondent groups. Lastly, hesitant respondents reported encountering cues of news about the advantages of COVID-19 vaccination significantly less often than non-hesitant respondents.

**Table 3 ijerph-19-03205-t003:** Between-Group Variance Analyses for Awareness Factors.

	Mean (SD)				F	*p*	Post-Hoc
	Total (*n* = 240)	Hesitant (*n* = 58)	Somewhat Hesitant (*n* = 69)	Non-Hesitant (*n* = 113)			
COVID vaccines can infect you with COVID (false)	0.84 (0.37)	0.71 (0.46)	0.91 (0.28)	0.87 (0.34)	5.75	0.00	H < S, N
Pregnant women are not advised to get vaccinated ^1^ (true)	0.55 (0.50)	0.62 (0.49)	0.61 (0.49)	0.47 (0.50)	2.57	0.08	n/a
COVID vaccines often have severe side-effects (false)	0.94 (0.24)	0.81 (0.40)	0.97 (0.17)	0.99 (0.10)	13.37	0.00	H < S, N
If enough people get vaccinated, the virus will spread less easily (true)	0.91 (0.28)	0.76 (0.43)	1.00 (0.00)	0.94 (0.24)	13.62	0.00	H < S, N
Getting vaccinated against COVID is more dangerous than getting infected with COVID (false)	0.97 (0.18)	0.86 (0.35)	1.00 (0.00)	1.00 (0.00)	14.38	0.00	H < S, N
COVID vaccines have been tested extensively (true)	0.81 (0.39)	0.47 (0.50)	0.88 (0.32)	0.95 (0.23)	40.88	0.00	H < S, N
COVID vaccines are deemed safe to use (true)	0.87 (0.34)	0.55 (0.50)	0.94 (0.24)	0.98 (0.13)	45.19	0.00	H < S, N
How high do you estimate your chance of getting COVID?	−0.17 (1.07)	−0.34 (1.10)	0.10 (1.03)	−0.25 (1.04)	3.41	0.04	n/a
How high do you estimate your chance of getting seriously sick of a COVID-infection?	−0.70 (1.27)	−0.76 (1.29)	−0.39 (1.29)	−0.86 (1.22)	3.06	0.05	S > N
How high do you estimate your chance of getting permanent damage from a COVID-infection?	−0.55 (1.24)	−0.55 (1.23)	−0.26 (1.21)	−0.73 (1.24)	3.06	0.05	S > N
How high do you estimate your chance of dying from a COVID-infection?	−1.66 (1.30)	−1.50 (1.22)	−1.72 (1.28)	−1.71 (1.35)	0.60	0.55	n/a
How bad would you mind getting COVID?	0.59 (1.26)	0.36 (1.28)	0.71 (1.26)	0.63 (1.23)	1.33	0.27	n/a
How bad would you mind getting seriously sick of a COVID-infection?	1.80 (1.00)	1.31 (1.27)	2.04 (0.74)	1.91 (0.90)	10.43	0.00	H < S, N
How bad would you mind getting permanent damage from a COVID-infection?	2.40 (0.75)	2.21 (0.93)	2.51 (0.59)	2.42 (0.72)	2.75	0.07	n/a
How bad would you mind dying from a COVID-infection?	2.61 (0.97)	2.33 (1.21)	2.75 (0.70)	2.67 (0.97)	3.49	0.03	H < S
I know people that got seriously sick of COVID	0.70 (0.46)	0.60 (0.49)	0.72 (0.45)	0.73 (0.45)	1.54	0.22	n/a
I got infected with COVID myself	0.17 (0.37)	0.26 (0.44)	0.14 (0.36)	0.13 (0.34)	2.37	0.10	n/a
I read a lot of news about the severity of COVID recently	0.73 (0.45)	0.60 (0.49)	0.77 (0.43)	0.77 (0.42)	3.10	0.05	n/a
I saw a lot of news about advantages of COVID vaccines recently	0.75 (0.43)	0.60 (0.49)	0.72 (0.45)	0.85 (0.36)	6.77	0.00	H < N
I saw a lot of news about dangers of COVID vaccines recently	0.64 (0.48)	0.69 (0.47)	0.67 (0.48)	0.60 (0.49)	0.77	0.46	n/a

^1^ when data was collected, this was still a correct answer. Since data collection, pregnant women have been cleared for vaccination.

### 3.5. Motivational Factors

As can be seen in Table 4, hesitant respondents were significantly less convinced of the emotional and rational advantages of COVID-19 vaccination, such as protection, hindering further spread and mutations of the virus, and returning to normal life. Hesitant participants reported significantly more negative rational and emotional disadvantages for COVID-19 vaccination than other respondents on the majority of attitude items. They believed less in the effectiveness of vaccines, the duration of protection, and were more afraid than non-hesitant and somewhat hesitant respondents of severe side effects. Furthermore, they expressed more negative feelings, such as anxiety and fear, towards vaccination. Somewhat hesitant and non-hesitant respondents mostly did not differ from each other on attitude items, apart from the emotional disadvantages, of which the non-hesitant respondents were significantly more convinced than the somewhat hesitant respondents.

As reported by the hesitant respondents, the social norms in their environment were more negative towards vaccination than for somewhat and non-hesitant respondents. Their partners, friends, family, and colleagues were all reported as less positive towards vaccination than those in the environment of the other two respondent groups. Hesitant respondents also expressed significantly lower levels of self-efficacy towards vaccination than the other two groups. Hesitant respondents had lower self-efficacy to get vaccinated than the other two groups regarding the obstacles of becoming sick of it, having to get two injections, finding it scary, having to go to the vaccination location alone, suffering from possible side-effects, having doubts about the effectiveness and duration of the protection, hearing contradicting information about vaccination, reading or hearing fake news about vaccination, getting told not to get vaccinated by people in their environment, and getting told that vaccination is pointless. Hesitant respondents had a significantly lower intention concerning the intention to get vaccinated with the AstraZeneca and Johnson & Johnson vaccine than somewhat hesitant and non-hesitant respondents.

**Table 4 ijerph-19-03205-t004:** Between-Group Variance Analyses for Motivational Factors and Intention.

	Mean (SD)				F	*p*	η^2^	Post-Hoc
	Total (*n* = 240)	Hesitant (*n* = 58)	Somewhat Hesitant (*n* = 69)	Non-Hesitant (*n* = 113)				
Attitude Pro Rational; If I get vaccinated against COVID-19 ^1^:								
It will protect people around me from getting COVID	1.57 (1.61)	0.78 (1.84)	1.81 (1.22)	1.83 (1.58)	10.05	0.00	0.08	H < S, N
It will protect me from getting COVID	1.74 (1.44)	1.03 (1.65)	2.07 (1.09)	1.90 (1.40)	10.25	0.00	0.08	H < S, N
It will help to hinder the further spread of COVID	1.88 (1.42)	0.98 (1.72)	2.12 (1.13)	2.20 (1.21)	17.58	0.00	0.13	H < S, N
Normal life will return sooner	1.87 (1.47)	0.79 (1.86)	2.12 (1.08)	2.27 (1.15)	24.87	0.00	0.17	H < S, N
I can go on a holiday again soon	1.26 (1.46)	0.57 (1.68)	1.30 (1.33)	1.58 (1.30)	10.00	0.00	0.08	H < S, N
It will contribute against the evolution of new virus variants	0.81 (1.79)	−0.12 (1.93)	0.90 (1.67)	1.24 (1.61)	12.27	0.00	0.09	H < S, N
The economy will be able to recover sooner	1.69 (1.30)	0.91 (1.74)	1.74 (1.00)	2.06 (1.01)	16.97	0.00	0.13	H < S, N
Attitude Pro Emotional; If I get vaccinated against COVID-19:								
I will feel safe	1.26 (1.45)	0.12 (1.84)	1.45 (1.09)	1.73 (1.05)	30.47	0.00	0.21	H < S, N
I will feel free again	1.17 (1.44)	0.16 (1.73)	1.43 (1.16)	1.52 (1.19)	22.24	0.00	0.16	H < S, N
I will feel protected	1.37 (1.37)	0.21 (1.68)	1.62 (0.89)	1.81 (1.07)	36.36	0.00	0.24	H < S, N
I will feel less alone	−0.14 (1.74)	−0.91 (1.81)	−0.19 (1.65)	0.28 (1.62)	9.84	0.00	0.08	H < S, N
I will be less worried about COVID	0.86 (1.55)	0.05 (1.80)	1.01 (1.41)	1.18 (1.34)	11.59	0.00	0.09	H < S, N
I will feel less guilty about breaking COVID regulations	−0.07 (1.88)	−0.55 (2.05)	−0.03 (1.78)	0.16 (1.82)	2.80	0.06	0.02	n/a
Attitude Con Rational; If I get vaccinated against COVID-19:								
It will not be effective	−1.78 (1.58)	−0.52 (1.96)	−2.04 (1.10)	−2.27 (1.24)	31.23	0.00	0.21	H > S, N
It will cost me too much time	−2.40 (0.82)	−1.81 (1.15)	−2.54 (0.58)	−2.62 (0.59)	23.31	0.00	0.16	H > S, N
I put myself at risk of getting COVID	−2.03 (1.29)	−1.10 (1.71)	−2.29 (0.77)	−2.35 (1.05)	23.71	0.00	0.17	H > S, N
I put myself at risk of getting severe side-effects	−1.02 (1.57)	0.14 (1.59)	−0.99 (1.44)	−1.64 (1.27)	30.74	0.00	0.21	H > S > N
I will predominantly support the pharmaceutical industry with it	−1.36 (1.65)	0.05 (2.03)	−1.70 (1.19)	−1.88 (1.22)	36.54	0.00	0.24	H > S, N
I do not know if that will end the pandemic	−0.42 (1.81)	0.74 (1.68)	−0.59 (1.67)	−0.92 (1.71)	19.04	0.00	0.14	H > S, N
Attitude Con Emotional; If I get vaccinated against COVID-19:								
It will feel unsafe	−1.60 (1.61)	−0.29 (1.96)	−1.65 (1.43)	2.24 (1.02)	36.33	0.00	0.24	H > S > N
It will feel scary	−1.43 (1.72)	−0.03 (2.11)	−1.45 (1.46)	−2.13 (1.11)	37.18	0.00	0.24	H > S > N
It will feel like I am gambling with my health	−1.39 (1.71)	0.29 (1.88)	−1.49 (1.39)	−2.19 (1.05)	61.77	0.00	0.34	H > S > N
It will make me feel nervous	−1.27 (1.73)	0.03 (1.88)	−1.22 (1.51)	−1.98 (1.34)	33.08	0.00	0.22	H > S > N
It will be because I feel forced to do it	−1.76 (1.75)	−0.14 (2.20)	−1.94 (1.44)	−2.48 (0.95)	48.86	0.00	0.29	H > S, N
Social norms: I should get vaccinated against COVID-19 according to my:								
My partner	1.27 (1.96)	−0.07 (2.25)	2.05 (1.30)	1.45 (1.82)	18.60	0.00	0.16	H < S, N
My friends	1.04 (1.73)	0.37 (1.84)	1.45 (1.29)	1.12 (1.82)	6.24	0.00	0.05	H < S, N
My parents	1.09 (1.77)	0.10 (1.96)	1.60 (1.41)	1.28 (1.68)	11.62	0.00	0.10	H < S, N
My family	1.13 (1.76)	0.28 (1.95)	1.54 (1.30)	1.32 (1.76)	9.06	0.00	0.08	H < S, N
My colleagues	0.93 (1.61)	0.49 (1.78)	1.51 (1.15)	0.82 (1.67)	5.70	0.00	0.06	H, N < S
My doctor	0.84 (1.78)	0.29 (1.63)	1.09 (1.53)	0.99 (1.94)	2.81	0.06	0.03	n/a
My religious leader	−0.90 (1.59)	−1.00 (1.44)	−1.04 (1.62)	−0.72 (1.72)	0.37	0.69	0.01	n/a
Self-efficacy: I would find it hard to get vaccinated against COVID-19:								
If it hurts	1.96 (1.36)	1.74 (1.66)	1.80 (1.38)	2.18 (1.14)	2.72	0.07	0.02	n/a
If it makes me sick	0.44 (1.82)	−0.48 (1.97)	0.36 (1.72)	0.96 (1.60)	13.54	0.00	0.10	H < S, N
If I have to get two injections	1.65 (1.56)	1.05 (2.00)	1.65 (1.38)	1.96 (1.31)	6.88	0.00	0.06	H < N
If I think it is scary	1.48 (1.72)	1.07 (2.02)	1.32 (1.63)	1.80 (1.55)	3.97	0.02	0.03	H < N
If I would have to go to the vaccination location alone	2.19 (1.06)	1.88 (1.37)	2.17 (0.95)	2.36 (0.90)	4.13	0.02	0.03	H < N
If I do not know which side effects it has	−0.08 (1.90)	−1.52 (1.50)	−0.22 (1.61)	0.73 (1.80)	34.77	0.00	0.23	H < S < N
If I do not know for sure if it will give me full protection	0.40 (1.86)	−0.91 (1.73)	0.61 (1.69)	0.96 (1.69)	23.85	0.00	0.17	H < S, N
If I do not know for how long it will protect me	0.23 (1.85)	−1.24 (1.48)	0.43 (1.63)	0.85 (1.74)	31.52	0.00	0.21	H < S, N
If I hear all kinds of contradicting information about the vaccine	0.07 (1.88)	−1.31 (1.52)	0.12 (1.65)	0.74 (1.80)	28.25	0.00	0.19	H < S < N
If I read or hear all kinds of fake news about the vaccine	0.68 (1.93)	−0.52 (1.78)	0.91 (1.78)	1.15 (1.85)	16.99	0.00	0.13	H < S, N
If people in my environment say I should not get vaccinated	1.42 (1.55)	0.93 (1.62)	1.54 (1.46)	1.60 (1.52)	3.98	0.02	0.03	H < N
If people in my environment say that vaccinations are pointless	1.50 (1.51)	0.81 (1.62)	1.62 (1.38)	1.79 (1.42)	8.93	0.00	0.07	H < S, N
Intention: Are you willing to get vaccinated against COVID-19:								
In general?	0.82 (0.47)	0.28 (0.72)	0.99 (0.12)	1.00 (0.00)	88.08	0.00	0.43	H < S, N
Within a year?	0.83 (0.45)	0.33 (0.71)	0.99 (0.12)	1.00 (0.00)	77.93	0.00	0.40	H < S, N
When you receive an invitation?	0.82 (0.48)	0.26 (0.72)	0.99 (0.12)	1.00 (0.00)	93.79	0.00	0.44	H < S, N
With the AstraZeneca-vaccine?	0.37 (0.72)	−0.50 (0.50)	0.07 (0.46)	1.00 (0.00)	385.78	0.00	0.77	H < S < N
With the Pfizer-vaccine?	0.82 (0.45)	0.28 (0.67)	0.99 (0.12)	1.00 (0.00)	101.34	0.00	0.46	H < S, N
With the Moderna-vaccine?	0.73 (0.53)	−0.03 (0.56)	0.91 (0.28)	1.00 (0.00)	224.96	0.00	0.66	H < S, N
With the Johnson & Johnson-vaccine?	0.53 (0.63)	−0.31 (0.47)	0.46 (0.50)	1.00 (0.00)	265.35	0.00	0.69	H < S < N

^1^ −3: completely disagree: +3 completely agree.

### 3.6. Regression

To identify how well the factors of the model explained the variance of the intention to get vaccinated, a multiple linear regression analysis was conducted (see Table 5). Inspection of the P-P plot, scatterplot of residuals, VIF, and tolerance suggested no multicollinearity. Linearity and homoscedasticity assumptions were met. All VIF values stayed between 1 and 5, indicating the absence of multicollinearity. Furthermore, all tolerance values stayed above 0.4, except for rational and emotional attitudes against vaccination in the last model, which were still higher than 0.2 [30]. 

In the first model, socio-demographic factors were entered, and it was revealed that people with paid work, a religion other than Catholicism, and lower educational level were more likely to be hesitant. Model 2 added the awareness factors of the ICM (knowledge, susceptibility, severity, and cues) to Model 1, revealing that lower levels of knowledge were uniquely associated with hesitancy. The socio-demographic factors were no longer significant, suggesting that their contributions were mediated by knowledge. Model 3 added the motivational factors of the ICM, revealing that emotional and rational disadvantages were uniquely associated with VH, whereas knowledge and positive emotional outcomes were positively associated with the intention to vaccinate. The final model accounted for 58 percent of the variance in vaccination intention.

**Table 5 ijerph-19-03205-t005:** Summary of the Stepwise Linear Regression for Intention to Vaccinate.

	B	SE B	β	Adj. R^2^	*p*
Model 1:				0.059	
-Working	−0.198	0.072	−0.178		0.006
-Religion other	−0.301	0.127	−0.153		0.019
-Education	0.060	0.027	0.143		0.028
Model 2:				0.332	
-Knowledge	0.215	0.023	0.546		0.000
Model 3:				0.577	
-Knowledge	0.060	0.023	0.153		0.009
-Attitude Con Emotional	−0.114	0.024	−0.338		0.000
-Attitude Con Rational	−0.111	0.036	−0.243		0.003
-Attitude Pro Emotional	0.053	0.024	0.129		0.027

## 4. Discussion

The first objective of this study was to determine possible differences in predisposing factors between hesitant, moderately hesitant, and non-hesitant individuals. Both the results of the chi-square analysis and the regression analysis suggest that in our sample, VH was more common among lower educated participants, which corresponds with findings from other studies regarding COVID-19 VH [31,32,33]. Previous research has indicated that religion can be associated with VH [34,35], because some religions may be against vaccination. In our sample, the majority was either not religious or Catholic, and we found VH more associated with having a religion other than Catholicism. This effect was also reported by other studies showing more hesitancy among individuals of Protestant faith and followers of anthroposophy [36]. However, we need to be cautious in the interpretation of this finding because of the relatively small sample size and the diversity within this group. Furthermore, previous research has indicated that people with a paid job were less likely to be hesitant [37,38], whereas our results suggest more hesitancy among those with a paid job. One explanation for this finding may be the relatively large proportion of students in this study (24.2%) who are highly educated but also less likely to have paid work. Finally, studies with a different culture and sample compositions reveal different outcomes concerning the importance of socio-demographic factors [39], thus illustrating the need for a culture specific approach to identify core socio-demographic drivers of COVID-19 VH. 

The second objective of this study was to examine possible differences between the groups in information factors and intentions to check information and source credibility. In line with prior findings, non-hesitant respondents made more use of written media than hesitant respondents [40]. Hesitant respondents used WhatsApp more often than the other two groups. Reliance on social media contributing to VH has also been reported by several other studies [41,42,43,44]. Additional research is needed to determine the mechanisms behind the information distribution through these sources and how they influence VH, possibly by second-hand misinforming [12]. A possible indication for this is that respondents indicated to use people in their surroundings as one of the most frequent sources for COVID-19 vaccination-related information. Previous research also indicated that health information on social media is of generally poorer quality in comparison to written media and might influence VH [45,46]. Furthermore, unregulated media (i.e., social media) are also more likely to contain misinformation and fake news, which could lead to higher VH in individuals with insufficient health literacy [46]. No significant differences between our three study groups were found for the intentions to check the quality of the information and to combat misinformation. This suggests that our respondents did not intend to undertake preparatory actions to check for fake news or source or news credibility. A possible explanation could be that exposure to misinformation on social media regarding COVID-19 vaccination may lead to an increase in information avoidance and heuristic processing [47,48]. Since the pandemic, many people may have been exposed to misinformation that has led to a new fake news pandemic. Hence, governments need to invest in strategies aimed at developing new algorithms to identify vaccine misinformation, as well as to implement and enforce bans on the content and advertising on their websites that contain misinformation [49]. Governments and public health organisations should actively focus on fostering informed decision-making by not only providing relevant and reliable information but also by encouraging people to check the quality of their information sources. A recent study in Germany found that higher fact-checkers more often used websites of public bodies to look up COVID-19 information and searched more often for specific COVID-19 information. Higher fact-checkers also had a higher age, more eHealth literacy, and believed less in common COVID-19 misinformation. Additionally, they had more knowledge about COVID-19, a more positive attitude towards fact-checking, and higher perceived social norms [50]. However, as perceptions of the reliability of official sources can be very low in certain subgroups when vaccination is at stake [23], this endeavour may not be easy. This may be especially challenging in the Netherlands, where the government opted for an early lowering of precautionary measures that resulted in steep increases of COVID-19 cases, which may have reduced the perceived reliability of governmental sources among the Dutch population. Lack of trust in authorities and pharmaceutical companies has also been reported in other studies [41,51,52]. Strengthening trust, however, is crucial [53].

The third objective of this study was to examine possible associations between awareness factors and COVID-19 VH. Significant differences were found between the hesitancy groups on almost all knowledge items. Hesitant respondents had less knowledge about COVID-19 and the vaccine than non-hesitant respondents. Knowledge was also found to have a unique association with intention in the linear regression analysis. Less knowledge about vaccination has previously been linked to increased VH for COVID-19 and other diseases [14,54], but not always for other vaccines [55]. A longitudinal study, also using the I-Change Model (ICM), showed that knowledge can serve as a distal factor for influencing attitudes [56]. This may also imply that having knowledge about relevant facts may be a first step, whereas becoming convinced of the relevance of these facts for one’s behaviour is a second step. Hesitant respondents indicated a significantly lower susceptibility than somewhat hesitant respondents for getting COVID-19, while prior research suggests that previous infection was related to a lower susceptibility, resulting in hesitancy [57]. Our hesitant group also reported a COVID-19 infection slightly more often than the other two groups, although these differences were not significant. Furthermore, hesitant respondents showed a lower perceived severity on the items about getting seriously sick of COVID-19 and dying of COVID-19 than the other respondents. Previous literature has also indicated that a lower perceived severity was linked to lower vaccine uptake [15,58]. An increase in perceived severity might therefore be a useful method to decrease VH. Lastly, hesitant respondents reported to perceive significantly less cues of news about the advantages of COVID-19 vaccination than non-hesitant respondents, which is also in line with previous research [29,57]. Providing more of these cues could be a possible strategy to combat VH in the future; however, more research may be needed to identify the most optimal source for providing these cues to avoid resistance. 

The fourth objective of this study was to research possible associations between motivational factors and COVID-19 VH. Significant differences were found between the groups on almost all attitude items, where hesitant individuals expressed a significantly lower attitude towards vaccination than other respondents. Similar trends were found in previous research [16,32,59,60]. Specifically, differences between groups were found on items that mentioned the possibility of severe side-effects or doubts about the effectiveness of the vaccination and the duration of the protection. The results of the regression analyses also supported their unique association with vaccine hesitancy and showed a unique relationship between emotional attitudes and vaccination intention. Other studies also reported that online negative arguments were found to be the main driver of VH [61] and that negative emotions influence COVID-19 VH [58]. An important feature of the ICM and our study is the distinction between emotional and rational outcomes. Emotional consequences were also found to be important for parents in their decision-making concerning the vaccination of their children [23]. Our findings thus suggest that emotional consequences (both advantages and disadvantages) played an important role in their attitude and intentions towards COVID-19 vaccination. Consequently, an implication for future campaigns is that the emotional feelings and related emotional consequences should be acknowledged more in (mass) media campaigns, as well as the need for highlighting the emotional advantages over the emotional disadvantages. Hence, while acknowledging potential emotional uncertainties, increasing the salience of emotional arguments for vaccination may thus be a possible strategy to decrease VH in the population. Furthermore, social norms in the environment of hesitant individuals were more negative regarding vaccination than for the less hesitant respondents. This is supported by previous research suggesting that a perceived negative social norm regarding vaccination can increase hesitancy [16,62]. Hesitant respondents also expressed a significantly lower self-efficacy towards vaccination, which is also congruent with previous publications [63,64]. This study adds detail to the specific self-efficacy-related situations that are relevant for hesitant Dutch adults. Although self-efficacy and social norms were not found to have a unique relation with VH in the regression analysis because of high correlations with attitude factors, the simple correlations with intention to vaccinate were 0.23 and 0.43, thus suggesting at least moderate importance. Future research is needed to identify their longitudinal impact. 

When assessing differences between groups on the intention items, hesitancy for either the AstraZeneca vaccine or the Johnson & Johnson vaccine was found to account for significant differences in the total VH score. This effect was expected due to the media coverage about the negative side effects of the latter two vaccines [65,66]. Pfizer emerged as the vaccine that people were the least hesitant about, which was also possibly caused by its positive media coverage compared to the AstraZeneca vaccine [67]. Shortly after the data collection of this study, the Netherlands also stopped vaccinations with AstraZeneca and Johnson & Johnson because of concerns over possible side effects.

A few limitations of this study should be considered when interpreting the present findings. First, due to the cross-sectional design, causation of the observed effects cannot be concluded. Future research should focus on these mechanisms, and also by employing longitudinal designs. Second, the research sample was predominantly female and included more highly educated individuals than nationally representative samples. This selection bias may be linked to the chosen recruitment channels and may have influenced the study outcomes, as a higher education has previously been linked to lower VH [31,32,33]. Research has also indicated that salient beliefs about VH may differ per specific group [68]. Hence, future research may consider purposive sampling through a variety of recruitment channels to reach particularly hesitant groups. Third, our study may be prone to participant bias. However, although the distributions of the three groups may not represent the true distributions at that time, our study purpose was to assess differences between these groups. More females than males participated in our study. However, the distribution of males and females did not differ for the three groups differing in their intention to become vaccinated. Fourth, this research is highly time sensitive. VH rates change and are also influenced due to the emergence and spread of new information on side effects and virus mutations. The threat of new and potentially more dangerous mutations may influence perceptions on VH rapidly as well as during times of lockdowns, which may increase individuals’ perceptions of risks and advantages of COVID-19 vaccination. 

## 5. Conclusions

Despite the limitations, the strength of this study is that it provides important insight into the psychosocial factors underlying COVID-19 VH. Using the ICM, a model that integrates various other models for understanding health behaviours [18], this study explained 58% of the variance in VH. This finding is similar to other studies on COVID-19 VH (e.g., [69]). Also in line with recent reviews [70], the findings reveal important differences between hesitant and non-hesitant individuals on various predisposing, motivational, awareness, and informational factors regarding COVID-19 VH. In terms of informational factors, no differences were found between groups for undertaking preparatory actions to check information credibility. However, differences in groups were found in the use of traditional media, where non-hesitant respondents used traditional written media more often than hesitant respondents and hesitant respondents used messaging services such as WhatsApp more often. In terms of predisposing factors, a higher educational level was associated with lower VH. Vaccine hesitant respondents had lower levels of vaccine knowledge, perceived less risks of COVID-19, and reported less cues to take action. They were less convinced of the rational and emotional advantages of vaccination, reported less favourable attitudes towards vaccination from others in their environment, and indicated lower self-efficacy to cope with barriers related to vaccination. Furthermore, a key finding of this study is the importance of emotional outcomes, which stresses the need to acknowledge perceptions of the emotional disadvantages in campaigns, while also increasing perceptions of the emotional advantages of COVID-19 vaccination, such as reducing feelings of regret, increasing feelings of freedom, and reassurance. Small pilot studies may be needed to analyse which communication strategies may be most successful at reaching groups that show high levels of VH in order to better understand how to increase their trust in the health information provided by official organisations and whether other more targeted approaches may be needed to better reach these groups. Finally, there will remain groups that are hesitant to become vaccinated against COVID-19. Several options may be required to reach these groups, including more personalized approaches. One strategy could be a mandatory vaccination requirement for the general public. Albarracin and colleagues found that such a strategy did not result in a backlash against vaccination in the United States [71]. Whether similar findings can be reported in reality and in other countries needs further analysis. 

## Figures and Tables

**Figure 1 ijerph-19-03205-f001:**
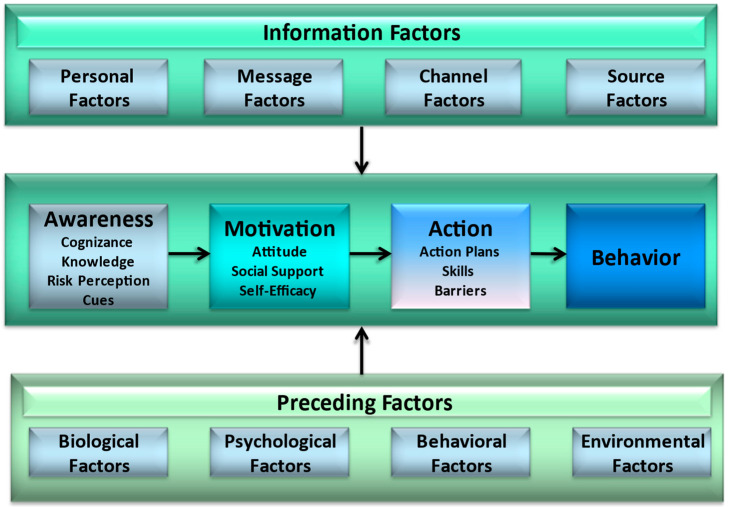
The I-Change Model [18].

**Table 1 ijerph-19-03205-t001:** Between-Group Comparison for Predisposing Factors.

	Total		Vaccine Hesitancy (VH)						Statistics		
	(*n* = 240)		Hesitant (*n* = 58)		Somewhat Hesitant (*n* = 69)		Non-Hesitant (*n* = 113)		χ^2^/F	df	*p*
Age (mean ± SD)	44.1 ± 17.2		46.3 ± 15.6		41.0 ± 15.9		44.9 ± 18.6		1.78	2	0.17
Gender	**%**	* **n** *	**%**	* **n** *	**%**	* **n** *	**%**	* **n** *	6.70	4	0.15
Male	25.8	62	21.0	12	18.8	13	59.7	37			
Female	73.8	177	26.0	46	31.6	56	42.4	75			
Other	0.4	1	0.0	0	0.0	0	0.9	1			
Education (mean ± SD)	6.50 ± 1.21		5.98 ± 1.42		6.80 ± 0.98		6.58 ± 1.14		8.09	2	0.00
Religion	**%**	* **n** *	**%**	* **n** *	**%**	* **n** *	**%**	* **n** *	8.43	10	0.08
No religion	53.8	129	17.8	23	31.0	40	51.2	66			
Catholic	39.6	95	30.5	29	24.2	23	45.3	43			
Other	6.6	16	37.5	6	37.5	6	25.0	4			
Paid work	**%**	* **n** *	**%**	* **n** *	**%**	* **n** *	**%**	* **n** *	6.22	2	0.05
Yes	72.1	173	27.2	47	30.6	53	42.2	73			
No	27.9	67	16.4	11	23.9	16	59.7	40			

**Table 2 ijerph-19-03205-t002:** Between-Group Variance Analyses for Information Seeking Behaviour.

How Often do You Use the Following News Sources for Information Regarding COVID-19 Vaccination? (Never-Very Often)	Mean (SD)				F	*p*	Post-Hoc
	Total (*n* = 240)	Hesitant (*n* = 58)	Somewhat Hesitant (*n* = 69)	Non-Hesitant (*n* = 113)			
Broadcast media (i.e., radio, television)	3.18 (1.23)	2.91 (1.22)	3.19 (1.33)	3.31 (1.17)	2.00	0.14	n/a
2 Written media (i.e., newspapers, journals)	2.02 (1.69)	1.50 (1.47)	2.13 (1.74)	2.21 (1.72)	3.71	0.03	H < N
3 Search engines (e.g., Google)	2.56 (1.43)	2.41 (1.46)	2.49 (1.48)	2.68 (1.37)	0.79	0.46	n/a
4 Online encyclopedias (e.g., Wikipedia)	0.88 (1.25)	1.02 (1.34)	0.97 (1.31)	0.76 (1.15)	1.05	0.35	n/a
5 Social media accounts or websites of public health organisations	2.51 (1.42)	2.28 (1.50)	2.39 (1.49)	2.71 (1.32)	2.15	0.12	n/a
6 Social media accounts or websites of official news media	3.05 (1.48)	2.66 (1.61)	3.10 (1.53)	3.21 (1.35)	2.84	0.06	n/a
7 Social media accounts or websites of other health organisations (e.g., health insurance, thuisarts.nl)	1.16 (1.26)	1.19 (1.34)	1.03 (1.20)	1.22 (1.27)	0.52	0.60	n/a
8 Other social media accounts or websites	1.50 (1.42)	1.78 (1.46)	1.39 (1.47)	1.42 (1.37)	1.49	0.23	n/a
9 Messaging services (e.g., WhatsApp)	1.45 (1.51)	1.88 (1.58)	1.28 (1.56)	1.35 (1.42)	3.12	0.05	n/a
10 YouTube	0.86 (1.27)	1.02 (1.31)	0.86 (1.34)	0.78 (1.20)	0.68	0.51	n/a
11 People surrounding you (i.e., family, friends)	3.03 (1.12)	2.98 (1.21)	3.07 (1.01)	3.02 (1.16)	0.10	0.90	n/a
12 Health professionals (e.g., your doctor, your GP)	1.45 (1.39)	1.55 (1.35)	1.58 (1.59)	1.31 (1.28)	1.03	0.36	n/a

## Data Availability

Data can be accessed upon request to the corresponding author (hein.devries@maastrichtuniversity.nl).

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
