# Peer review of "COVID-19 Vaccine Hesitancy: The Role of Information Sources and Beliefs in Dutch Adults"

_ijerph, 2022, doi:10.3390/ijerph19063205_

Round 1

Reviewer 1 Report

In overall, the following seems to me the most salient positive points;  Bibliographic references; Significance of the issue and Presentation of the findings and results.

The weakness points are the cross sectional aspect, the selection bias of participants and the weak analysis of study limitations

I suggest the following improvements:

  • to revise the summary by highligting the results in accordance with the four study objectives
  • to provide information on the period of the study ( epidemiological context, government strategy, degree of public crisis, pressure from the medical sector...)
  • to be more explicit on the participant bias. ( including the difference between the selected group and others-the vaccinated one)
  • to develop the section on the study strengths and limitations not only from a methodology point of view but also from an operational and strategic public health perpective.

Author Response

The weakness points are the cross sectional aspect, the selection bias of participants and the weak analysis of study limitations. I suggest the following improvements:

  1. to revise the summary by highlighting the results in accordance with the four study objectives

Response: We thank the reviewer for this observation and we changed this. The results section is now:

Results: Vaccine hesitant participants (n = 58) had lower levels of education, more often paid work, and tended to have a religion other than Catholicism. They used written media less often and tended to use less often website of public health organization, but used messaging services like WhatsApp more often. All participants had similar neutral intentions towards checking information credibility. They had less knowledge about vaccination, lower perceived severity of getting sick and dying of COVID-19, and reported fewer exposure to cues about advantages of COVID-19 vaccination. They were less convinced of the emotional and rational advantages of COVID-19 vaccination and expressed more negative feelings about it. The reported more negative social norms concerning COVID-19 vaccination, and lower self-efficacy to get vaccinated and to cope with potential side-effects.

  1. to provide information on the period of the study ( epidemiological context, government strategy, degree of public crisis, pressure from the medical sector...).

Response: We added additional information. However, we do not elaborate on the degree of public crises and pressure from the medical sector, as different views on these topics exist and interpretation of these items can be subject to (personal) bias.

This study used a convenience sample and focused on Dutch-speaking adults, aged 18+, as they were the target population for the vaccination campaign against COVID-19 in the Netherlands in which four types of vaccinations were used (Pfizer, Moderna, AstraZenica, Johnson & Johnson), allocation of vaccines was decided top-down depending on the age of the person.

  1. To be more explicit on the participant bias. ( including the difference between the selected group and others-the vaccinated one)

Response: We understand this remark, and this is always a concern with these types of studies. Our purpose, was to understand the differences between the three groups. We added the following remark in the discussion:

Third, our study may be prone to participant bias. However, although the distributions of the three groups may not represent the true distributions at that time, our study purpose was to assess differences between these groups. More females than males participated in our study. However, the distribution of males and females did not differ for the three groups differing in intentions to become vaccinated.

  1. To develop the section on the study strengths and limitations not only from a methodology point of view but also from an operational and strategic public health perspective.

Response:

We decided to describe the limitations and not the strengths in the limitation section. But indeed, the reviewer is right, we did not mention the strengths explicitly. Hence we altered the intro of the final conclusion.

Despite the limitations, the strength of this study is that it provides important insight into the psychosocial factors underlying COVID-19 VH.

Response: Concerning the strategic issues, we added the following:

Small pilots may be needed to analyse which communication strategies may be most successful at reaching groups that show high levels of VH in order to better understand how to increase their trust in the health information provided by official organizations, and whether other more targeted approaches may be needed to better reach these groups. 

Reviewer 2 Report

I read the manuscript by De Vries et al. and found it interesting and topical. However, I think there are some changes to be made before publication.

Title: 
Why not call the phenomenon Vaccine Hesitancy as properly addressed throughout the rest of the manuscript? (same thing in 2.3.1)

Abstract: 
L16: Need to specify that the sample is of convenience.
L18: I would report the percentage instead of the absolute number
I would add in the methods the fact that the I-Change Model was used.

Introduction:
L64-65 There are more relevant sources of determinants of VH. Both the SAGE working group and the WHO BeSD model are fine.
From L70 onwards, I think this section should be moved to Methods.

Results:
L209 There are differences between genders. This should be reported and also discussed.

Discussion:
I find the section well done and the boundaries correctly addressed.

Conclusion:
I recommend streamlining the section. I think, for example, that the last few sentences relate more to a discussion not being the result of this manuscript.

Author Response

I read the manuscript by De Vries et al. and found it interesting and topical. However, I think there are some changes to be made before publication.

Title: 
1. Why not call the phenomenon Vaccine Hesitancy as properly addressed throughout the rest of the manuscript? (same thing in 2.3.1).

Response: Good point, thanks! We changed the title, also made changes in 2.3.1 and in the discussion.

  1. Abstract:
    L16: Need to specify that the sample is of convenience.
    L18: I would report the percentage instead of the absolute number
    I would add in the methods the fact that the I-Change Model was used.

Response: L16. Thank you for mentioning this. We added the that the sample is of convenience, added the percentage to the absolute number. The use of the ICM is already mentioned in the abstract, so we did not want to repeat it.

A cross-sectional online survey using a convenience sample was conducted among Dutch adults (N = 240).

  1. Introduction:
    L64-65 There are more relevant sources of determinants of VH. Both the SAGE working group and the WHO BeSD model are fine.
    From L70 onwards, I think this section should be moved to Methods.’

Response: There are indeed many more sources. We opted for a selection of them in order to avoid a too long list of references (which was initially at least twice as long as it is now). We believe that we cover the most important ones relevant for this study so we decided not to add specific other studies, unless the reviewer believes that we have omitted a very important study. Second, sometimes the discussion of the theoretical background is moved to the method to indicate the theoretical underpinning of the study. However, we also discuss what is currently known to better introduce the rationale for our study. Hence, we believe that this section should be placed in the introduction.

  1. Results:
    L209 There are differences between genders. This should be reported and also discussed.

Response: The reviewer correct, we indeed attracted more female respondents. As can be seen this did not result in differences concerning the distribution of the three groups. We now added the following section in the limitations of the study. Thanks for the observation as we did not mention this indeed!

Third, our study may be prone to participant bias. However, although the distributions of the three groups may not represent the true distributions at that time, our study purpose was to assess differences between these groups. More females than males participated in our study. However, the distribution of males and females did not differ for the three groups differing in intentions to become vaccinated.

  1. Discussion:
    I find the section well done and the boundaries correctly addressed.

Response: Thank you very much!

  1. Conclusion:
    I recommend streamlining the section. I think, for example, that the last few sentences relate more to a discussion not being the result of this manuscript.

Response: We altered the introduction of this section slightly due to a comment of the other reviewer. Concerning the last point we understand the comment. At the same time we think that it is important to reflect on potential pathways to be considered for public health on how to best reach groups for public health issues, also those that are difficult to reach. Hence, we feel that this section should stay as it is.